# AI-Driven fetal distress monitoring SDN-IoMT networks

Amin Ullah[1], Qazi Mazhar Ul Haq[1]*, Zabeeh Ullah[2],
Jaroslav Frnda[3,4], Muhammad Shahid Anwar[5]*

1 Department of Computer Science and Engineering and International Bachelor Program in Informatics, Yuan Ze University, Zhongli, Taiwan, 2 Department of Computer Software Engineering, National University of Sciences and Technology (NUST), Islamabad, Pakistan, 3 Department of Quantitative Methods and Economic Informatics, Faculty of Operation and Economics of Transport and Communication, University of Zilina, Zilina, Slovakia, 4 Department of Telecommunications, Faculty of Electrical Engineering and Computer Science, VSB-Technical University of Ostrava, Ostrava, Czech Republic, 5 Department of AI and Software, Gachon University Seongnam-si, Seongnam-si, South Korea

* qazi@saturn.yzu.edu.tw (QMUH); shahidanwar786@gachon.ac.kr (MSA)

**Data availability statement:** We have uploaded our minimal dataset to the public repository

## Abstract

The healthcare industry is transforming with the integration of the Internet of Medical Things (IoMT) with AI-powered networks for improved clinical connectivity and advanced monitoring capabilities. However, IoMT devices struggle with traditional network infrastructure due to complexity and eterogeneous. Software-defined networking (SDN) is a powerful solution for efficiently managing and controlling IoMT. Additionally, the integration of artificial intelligence such as Deep Learning (DL) algorithms brings intelligence and decision-making capabilities to SDN-IoMT systems. This study focuses on solving the serious problem of information imbalance in cardiotocography (CTG) characteristics with clinical data of pregnant women, especially fetal heart rate (FHR) and deceleration. To improve the performance of prenatal monitoring, this study proposes a framework using Generative Adversarial Networks (GAN), an advanced DL technique, with an auto-encoder model. FHR and deceleration are important markers in CTG monitoring, which are important for assessing fetal health and preventing complications or death. The proposed framework solves the data imbalance problem using reconstruction error and Wasserstein distance-based GANs. The performance of the model is assessed through simulations performed using Mininet, according to criteria such as accuracy, recall, precision and F1 score. The proposed framework outperforms both the basic and advanced DL models and achieves an effective accuracy of 94.2% and an F1 score of 21.1% in very small classes. Validation using the CTU-UHB dataset confirms the significance compared to state-of-the-art solutions for handling unbalanced CTG data. These findings highlight the potential of AI and SDN-based IoMT to improve prenatal outcomes.

## Introduction

The medical industry faces challenges in adopting emerging technologies, yet advancements like extended reality (XR), Metaverse, and IoT big data analytics have the potential to

"Figshare" to ensure transparency and reproducibility of our research. The datasets are accessible via the following links: Training dataset: https://figshare.com/s/33421e16d980a6cca55c. Testing dataset: https://figshare.com/s/1dd8a4372a4580173569.

**Funding:** This research was supported by the institutional research of the Faculty of Operation and Economics of Transport and Communications—University of Zilina, no. 3/FPEDAS/2024.

**Competing interests:** The authors have declared that no competing interests exist.

revolutionize healthcare [1–3]. These technologies enable advanced services such as telepresence, surgery, and smart diagnostics through IoT-enabled devices [4]. With the integration of artificial intelligence (AI) and DL models, networks can process large amounts of medical imaging data in real-time, improving diagnostic accuracy and personalized treatment. These developments are poised to enhance IoT device performance and lead healthcare into a smarter, more connected future [5]. One of the fastest-growing segments within the IoT market is the IoMT, also known as connected devices in healthcare. These devices, such as wearable fitness trackers, smart medical implants, and remote patient monitoring systems, have transformed the medical field [6,7]. They enable continuous patient monitoring, allowing medical personnel to collect vital patient data and make informed decisions [8–10].

However, vulnerabilities in IoT infrastructure pose significant challenges, especially as networked devices become targets cyberattackscks [11]. Issues like scalability, heterogeneous, and interoperability are prevalent due to the diverse array of IoT devices that must communicate effectively to meet growing demands [12,13]. Strong legal frameworks are also required to address data privacy concerns, given the large volumes of personal information these devices capture. SDN in IoT environments offers a solution, focusing on data integrity and computational time complexity [14,15]. SDN is crucial for managing the unique challenges of IoT in healthcare by providing dynamic network management and prioritized data transfer for essential medical equipment. This enhances the reliability of telemedicine and patient monitoring systems. SDN's robust security features help protect sensitive medical data, maintain HIPAA compliance, and mitigate cyber threats, making it essential for the secure operation of IoT devices in healthcare [16].

The integration of SDN with IoMT, or SDN-IoMT, further improves healthcare infrastructure. By leveraging deep learning, SDN-IoT can analyze vast amounts of data generated by IoT devices in real-time, enabling early disease detection, personalized treatment recommendations, and predictive analytics. Additionally, DL can optimize network resource allocation, ensuring efficient data transfer and reduced latency for critical healthcare applications. This approach has the potential to improve patient care quality, streamline healthcare operations, and support proactive, data-driven healthcare solutions [17]. The application of DL models for anomaly detection and classification in SDN-IoT medical contexts is shown in Fig 1. The acronyms used in this article are shown in Table 1.

In medical settings, deep learning models are being used for anomaly detection and classification within SDN-IoT environments, significantly enhancing medical care. For example, CTG is a widely used technique to continuously monitor fetal heart activity during high-risk pregnancies [18]. Despite its effectiveness, the subjective nature of visual interpretation can lead to variability in diagnoses. Computerized CTG, powered by deep learning, addresses this issue by providing accurate, real-time analysis of fetal heart rate patterns, helping clinicians prevent fetal morbidity and mortality through timely interventions [19] and shown in Fig 2.

FHR, deceleration, and uterine contractions are the three key characteristics of CTG that are predominantly discussed in this article, along with other medical measures like blood pressure, temperature, and oxygen levels that are related to the health of mothers. These metrics are crucial for keeping an eye on fetal distress. The most crucial measures are FHR, deceleration, and uterine contractions since any unsettling aspect of either calls for quick action from medical professionals. Fig 3 displays FHR, BV, and acceleration. Nevertheless, difficulties occur when DL algorithms are used to identify and categorize anomalies for fetal distress monitoring on an unbalanced dataset. The dataset is gathered in a dispersed setting with widespread wireless connection from a variety of medical sensors, Internet of Things devices, and CTG machines. In an unbalanced dataset, the minority of cases entail malformations, but the majority of cases show normal behavior and usually suggest normal fetal health.

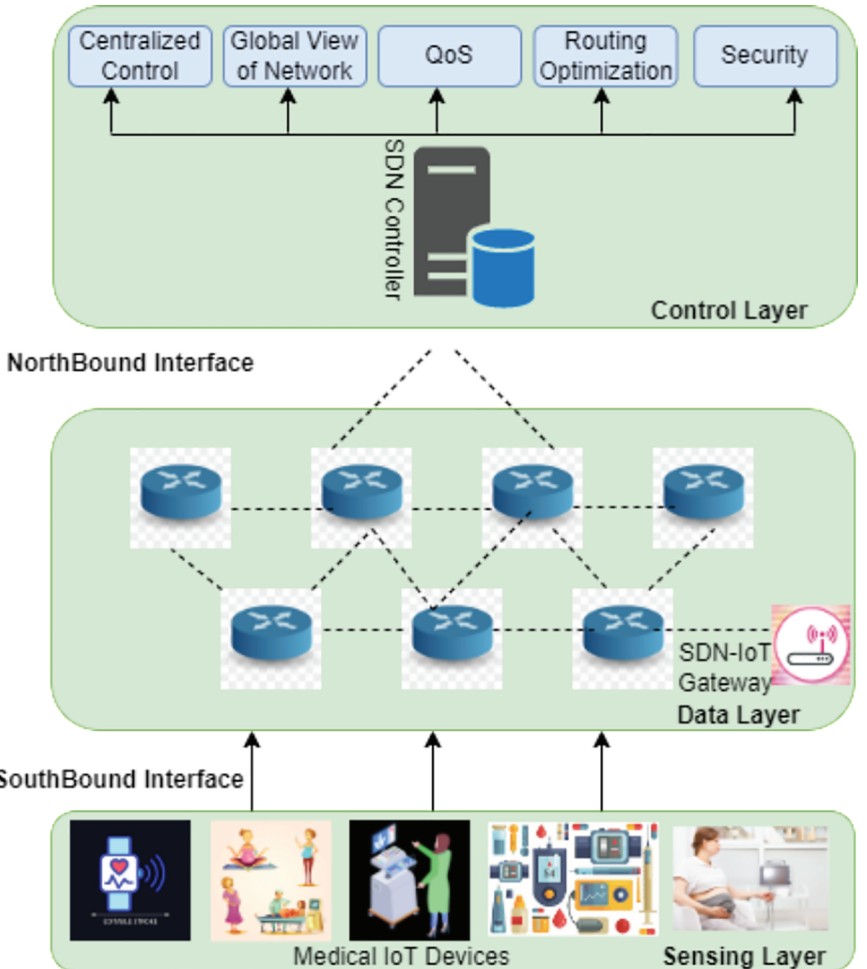

**Fig 1. The sensing layer gathers data and sends it to the data layer, while the control layer acts as the infrastructure's central intelligence, overseeing the whole SDN-IoT network.** An example of an SDN-IoMT infrastructure consists of these three layers.

**Table 1. A description of CTG and clinical features, including the types of sensors used to capture these features and the normal range for each feature.**

| Sr. No. | Feature Name | Sensor Type | Normal Range |
|---------|--------------|-------------|--------------|
| 1 | Maternal Blood Pressure | B.P Sensor | $\leq$ 139/89 |
| 2 | Maternal Heart Rate | Heart-beat Sensor | < 90/min |
| 3 | Maternal Oxygen Saturation (%) | Oxygen Level Sensor | $\geq$ 90% |
| 4 | Maternal Temperature | Temp. Sensor | 97–99 F (36.1–37.2 °C) |
| 5 | Fetal Heart Rate | Fetal Heart Rate Sensor | 110–160 |
| 6 | Baseline Variability | Baseline Variability Sensor | 5–25 |
| 7 | Acceleration | Acceleration Sensor | 2 Acc. per 20 minutes |
| 8 | Deceleration | Deceleration Sensor | 0 |

Only a few particular and exceedingly rare types of abnormalities stand out among these rare occurrences of fetal distress; most are well known. Because of this imbalance, the model may become biased toward the majority class and find it difficult to train and detect rare or minority class abnormalities. This presents issues when designing machine learning or deep learning

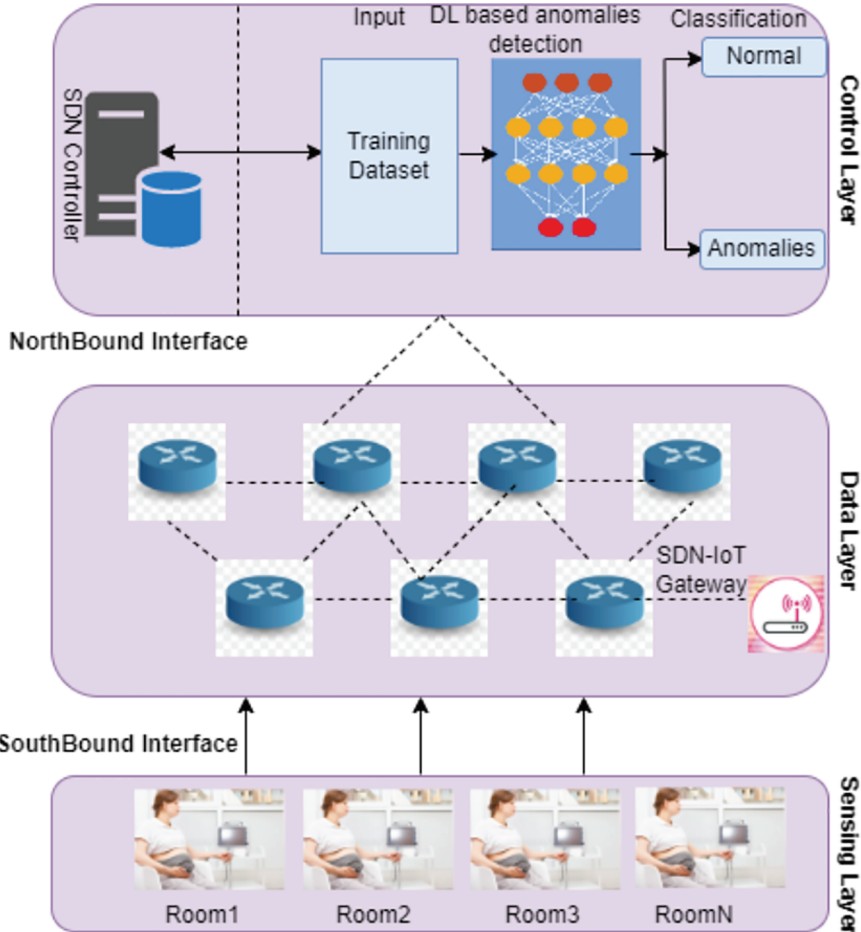

**Fig 2. To monitor fetal distress monitoring, many CTG devices are placed in rooms within a distributed SDN-IoT architecture.** After this, the control layer receives the data gathered from these devices for additional processing. We apply a Deep Learning model in this three-layered architecture to identify and categorize anomalies

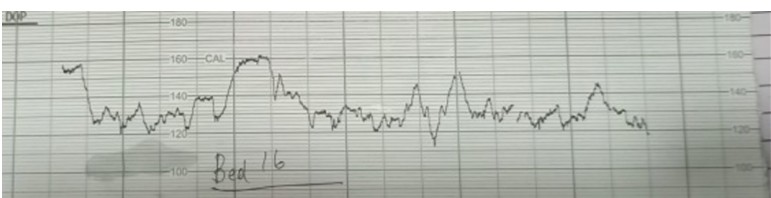

**Fig 3. An actual CTG recording was acquired from a Pakistani hospital located in Islamabad.** Three important data are shown on this CTG: acceleration, baseline variability, and fetal heart rate. All three are within the normal range.

algorithms for CTG analysis. We have presented a system based on GAN and Autoencoders to address the data imbalance problem in CTG features. The GAN in this system is in charge of producing artificial CTG feature data. Specifically, we study the Wasserstein distance-based GAN and a version of GAN called the Reconstruction Error [20]. This version of the GAN can provide plausible generated synthetic data for small anomaly flows.

Our suggested framework's design is shown in Fig 4, and it consists of four different layers: the Application Layer, the Data Layer, the Control Layer, and the Sensing Layer. Pregnant women's vital information is gathered by a wide range of medical IoT devices that make up the Sensing Layer. The second component is the Data Layer, which is made up of diverse networking devices that allow data to be seamlessly transferred across the layers. In these connections, we take it that the OpenFlow protocol is being used. Aptly titled the Control Layer, the third layer of our architecture is sometimes referred to as the 'brain' of the system. In

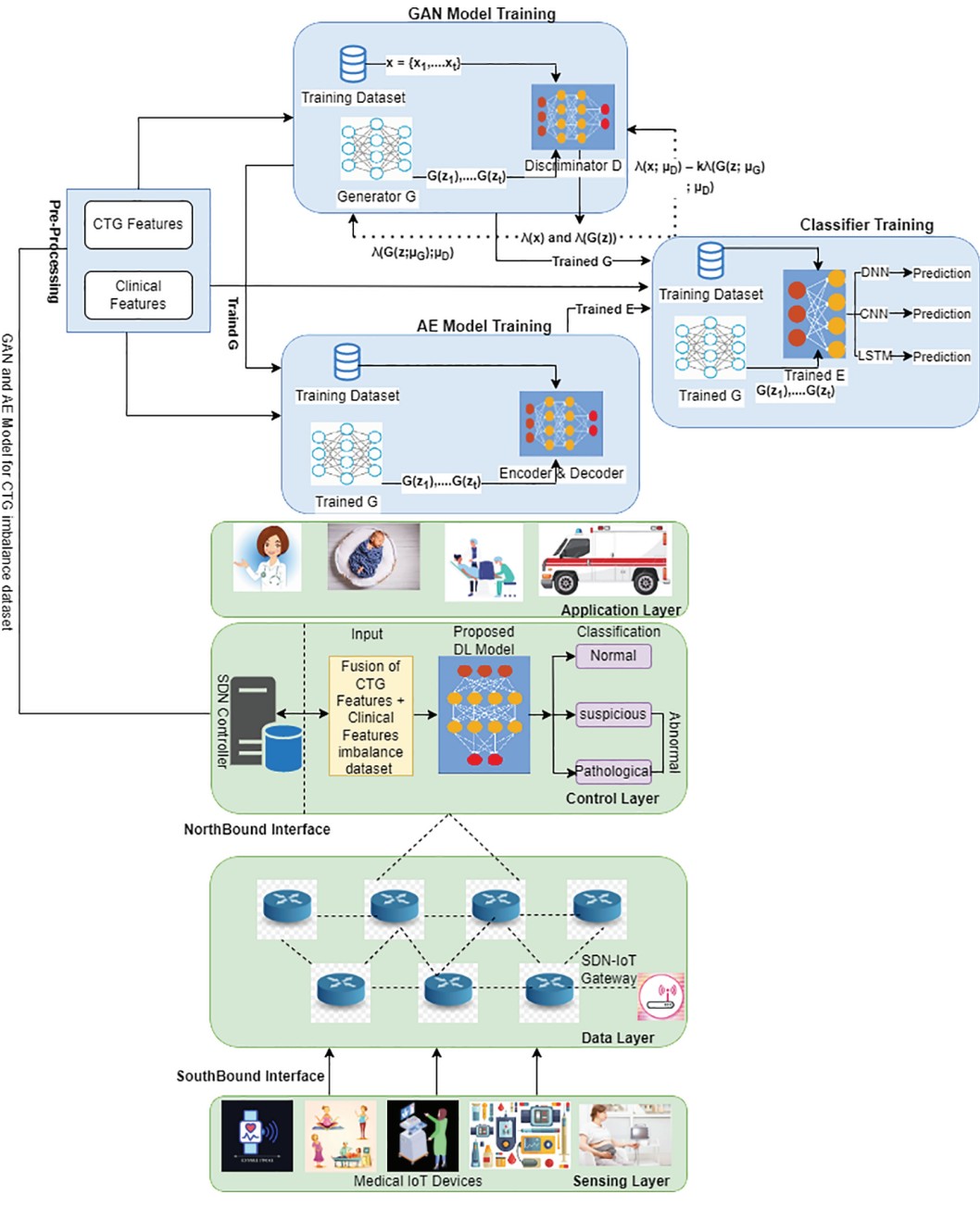

**Fig 4. The architecture of the proposed framework.**

this case, the SDN controller effectively governs data communication by providing the Data Layer with rules based on its global view of the entire underlying network. Notably, as illustrated in Fig 5, we apply the Generative Adversarial Network-Autoencoder (GAN-AE) deep learning technique within this Control Layer, which is intended for the identification and categorization of abnormalities within unbalanced CTG datasets. Data pre-processing, GAN model training, AE model training, and predictive model training are the four main parts of the GAN-AE deep learning model. Raw datasets are converted into a structured format that is suitable with deep learning models during the first stage of data pre-processing. The GAN model is then trained, and it is crucial to the training of the AE model that comes after. Training the Predictive model, which gains from the trained AE and GAN models, is the last stage. Here, the AE serves as a feature extractor inside the model, and the GAN is used to produce sparse data. We have used three different deep learning models—the Long Short-Term Memory (LSTM), the Convolutional Neural Network, and the Deep Neural Network—to complete classification tasks. Based on the information gathered from the lower layers, gynecologists make decisions about future steps pertaining to fetal health at the fourth tier, known as the application layer. We tested the suggested framework using an unbalanced dataset that we obtained from an Islamabad public hospital in order to evaluate its performance. Our investigations show that the performance of our suggested system outperforms a lot of naive findings from DL models. Additionally, by resolving the problem of data imbalance, the performance of current DL-based CTG anomaly detection and classification models can be improved.

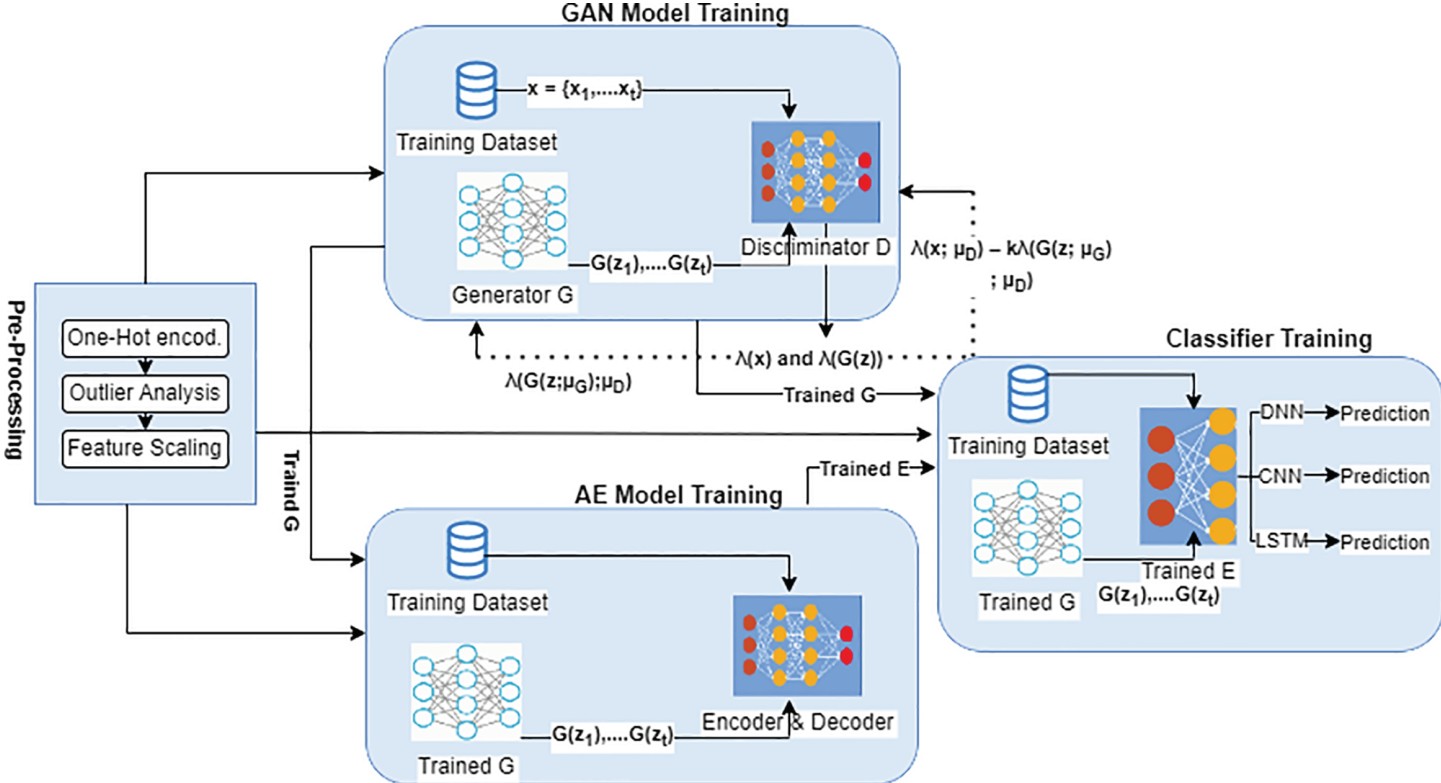

**Fig 5. Creation of a model for tracking fetal health monitoring utilizing an imbalanced CTG dataset, Generative Adversarial Network, and Autoencoder for anomaly detection and classification.**

The major contributions of our article can be summarized as follows

- To reduce the obstacles associated with IoT, such as heterogeneous, complexity, and security, we have created an SDN-based Medical IoT architecture. To tackle the issue of imbalanced CTG features and clinical data, we have presented a unique GAN-AE-based system in the context of an SDN-IoT environment.To completely examine the proposed model, its performance has been compared with baseline methodologies, including naive DL models and advanced DL models. Every model was evaluated and trained in the same setting to guarantee an equitable comparison.
- The main objective is to improve the monitoring of fetal distress by precisely identifying and categorizing small class abnormalities. To overcome the imbalance in CTG feature data, we utilize the Wasserstein distance-based GAN and reconstruction error inside this framework to produce believable synthetic data.
- To evaluate the performance of our proposed method, we created a dataset with 8 features (42,000 instances) that included FHR and deceleration. Additionally, the CTU-UHB dataset, a publicly available CTG dataset, is used to train the proposed model and evaluates its performance against the most recent methods found in the literature.

This is how the rest of the article is structured: Sect 2 provides an overview of the relevant work from the standpoint of machine learning and deep learning algorithms for anomaly detection and classification in the CTG dataset. We go into great depth about our four-layered methodology and the stages of the proposed anomaly detection and classification model in Sect 3. In Sect 4, the proposed system is assessed using an imbalanced CTG dataset, and comprehensive analysis of the experimental findings is provided. Sect 5 presents our analysis and discussion of the obtained outcomes. In Sect 6, we finally provide the study's closing thoughts.

## 1 Related work

The recent research on machine learning (ML) and DL techniques for identifying and categorizing anomalies in fetal distress monitoring using imbalanced CTG data will be reviewed in this part. The authors of [21] presented a TS-GAN-based model designed to address the issue of data imbalance in CTG for fetal health monitoring. To increase the amount of data, they also used augmentation based on GANs. The classification results from enhanced GAN were shown by the authors to be superior to those from non-augmented classification. The authors of [22] presented an inventive method called IBTF. Using the CTU-UHB open-access CTG dataset, they tested the model and found that it was 64.01% accurate in identifying anomalies within the minor classes. The authors of [23] presented a deep learning-based approach with the goal of increasing hypoxia detection accuracy. To improve the overall accuracy, they improved a number of steps, such as input representation, data scaling, and classifier layer modifications. The findings showed that the suggested model has a 50.21% accuracy rate. Using the publicly available CTU-UHB CTG dataset, [24] evaluated deep learning models (DenseNet, CNN) with ensemble learning models (SVM, Decision Tree) for hypoxia detection. The accuracy of the suggested methods was 60.91% and 65.91%, respectively. To find anomalies in an unbalanced CTG dataset, the authors of the paper [25] examined a number of machine learning techniques, such as SVM, Random Forest (RF), and K-nearest neighbors. With an accuracy of 90.8%, the results showed that the RF algorithm performed better than any other ML method. In [26], authors proposed a multimodal deep learning architecture (MMDLA) for intelligent antepartum fetal monitoring that is capable of performing automatic CTG feature extraction, fusion with clinical data and classification. The data imbalance issue was also effectively resolved by the LGBM classifier With an accuracy of 90.77%.

In [27], a Multi-Layer Perceptron (MLP) and Convolutional Neural Network (CNN) based approach were proposed for fetal distress monitoring. The method achieved an accuracy of 80.1% when evaluated on a clinical dataset. Another approach, presented by [28], employed a Support Vector Machine (SVM) based technique named Alexnet for the precise classification of CTG datasets. This model computed the time complexity during the classification process and attained an accuracy of 93.2% on publicly available CTG data. In [29], they conducted a comparative analysis among various machine learning and deep learning based classification methods applied to CTG datasets. The primary aim of their study was to delineate the boundaries of different classification algorithms and contrast their predictive accuracies to identify the best classifier for assessing fetal well-being. In [30], they proposed the use of seven machine learning methods to classify fetal health into three states: normal, suspect, and pathological. Among these methods, GCB demonstrated superior performance, achieving a classification accuracy of 93.9%. [31], a Single Layer Perceptron based approach was proposed for accurate CTG data classification and fetal state prediction. Their method, evaluated on a CTG dataset, achieved an impressive classification accuracy of 94.0%. [32] developed a multimodal bidirectional gated recurrent units (MBiGRU) network for end-to-end CTG feature extraction and classification. The effectiveness of the proposed MBiGRU model was tested on 16,355 antenatal CTG records collected from collaborating hospitals, with consistent case interpretation by three expert obstetricians. The experimental results, employing ten-fold cross-validation, demonstrated an average accuracy of 86.45%. [33] presented the framework uses an ensemble of feature extraction techniques after extracting features from the ECG signal using a real-time electrocardiogram-cardiotocography (ECG-CTG) device. A dataset of ECG signals is used to evaluate the suggested framework, and the findings demonstrate that it is capable of accurately and efficiently classifying ECG signals with minimal computing overhead. In [34], the authors experimented with a variety of categorization models and found that, when compared to both deep learning and conventional machine learning models, data augmentation clearly outperforms both in terms of performance measures. The purpose of this study [35] is to present several AI techniques, each of which serves as a helpful guide to help overcome the current difficulties with CTG and forecast fetal problems. In the first phase of the study, we denoised CTG signals and used an algorithm to extract significant features in accordance with guidelines from the Japan Society of Obstetricians and Gynecologists (JSOG). We then tested the algorithm's ability to detect high-risk births on a large dataset collected under clinical conditions by applying four machine learning techniques to the extracted features: SVM, RF, DT, and ANN.

The literature review highlights several challenges in achieving optimized accuracy for imbalanced datasets. Deep learning models often struggle to effectively learn the behavior of imbalanced data, leading to a bias toward majority instances. Additionally, the selection of CTG and clinical features significantly influences fetal distress monitoring. However, many studies in the literature have focused solely on CTG features, neglecting the inclusion of crucial clinical features. Furthermore, a majority of the literature relies on manual methods for extracting CTG features, resulting in poor generalization. Moreover, when selecting clinical features in a distributed environment using IoT devices, the complexity and heterogeneous of these devices often lead to a deterioration in accuracy. To the best of our knowledge from the reviewed literature, there has been no utilization of SDN to manage IoT heterogeneous and complexity.

Hence, this article aims to attain optimized accuracy by integrating CTG and clinical features. Furthermore, the study introduces SDN to mitigate IoT heterogeneous and complexity. Additionally, it employs a deep learning-based automatic feature extraction method to augment the generalization of the proposed model.

## 2 Proposed framework

Fig 5 illustrates the four layers that make up our proposed structure. Layer 1 is for sensing; layer 2 is for data; and layer 3 is for control. We have included the GAN-AE model for identifying and categorizing abnormalities in CTG imbalance data within the third layer, the Control Layer. We give a thorough explanation of every layer and module in the suggested framework in this section.

### 2.1 Sensing layer

The core component of our suggested paradigm is the Sensing Layer, which includes a range of Medical IoT (MIoT) gadgets, including actuators and sensors. These gadgets include sensors for detecting CTG properties like FHR and BV, as well as sensors for blood pressure, temperature, and oxygen saturation. This layer's main responsibility is to collect data and forward it to the next layer. The Sensing Layer is vulnerable to a variety of security threats because of the large diversity and heterogeneous of its sensors. It is imperative to guarantee the precise and dependable gathering of data together with its smooth progression to the subsequent tier.

### 2.2 Data layer

The data layer is the second layer in the suggested model. This layer includes a variety of networking hardware, such as switches and routers with SDN support. This layer's main job is to make it easier for data to transfer across switches by following the instructions and guidelines that come from the control layer. As a result, the data layer is frequently called the "dumb layer." The South Bound Interface (SBI) facilitates communication between the control layer and the data layer.

### 2.3 Control layer

The Control Layer, the third layer in our suggested architecture, acts as the system's main control center. This layer's SDN controller has a comprehensive view of the entire network and controls every aspect of it, including efficiently and effectively handling congestion, routing optimization, Quality of Service (QoS), and security enhancement. A range of DL algorithms are integrated into the SDN controller to further improve its efficacy, efficiency, and intelligence. To identify and categorize abnormalities in CTG imbalance datasets, we have incorporated the GAN-AE deep learning model into the SDN controller in our framework. Below is a thorough explanation of every module in the GAN-AE system:

**2.3.1 Data pre-processing.** At this point, unprocessed data is cleaned up and formatted so that DL algorithms may use it. As seen in Fig 6, we processed the raw CTG signals using the preprocessing method outlined in [26]. Three essential processes are involved in this process: segmentation, standardization, and interpolation. First, outliers are eliminated from the start and finish of the signals. Nan is used instead of interpolation unless there are sudden oscillations in the signal. For instance, Nan is given any untrustworthy values found in the FHR analysis. Assuming that $1 < st < N$, $d_{st}$ indicates the first value inside an outlier or missing segment. After that, we pick the segment $d_{1:st}$ and compute its median value, which we refer to as $d_{med}$. This value is then used to substitute $d_{st}$. Baseline Variability (BV) and FHR signals also synchronize, which prevents these signals from being removed or interpolated using FHR data. When a missing segment is indicated by the notation $b_{st}$, where $1 < st < N$, we can choose segment $b_{st}$ and replace it with $b_{med}$. The remaining missing portions are addressed using this

**Fig 6. The pre-processing of CTG raw signals [26].**

method consistently. Furthermore, the same technology as previously stated is used to process acceleration and deceleration, which are crucial CTG signals in fetal distress monitoring. Below is an outline of FHR's decentralization:

$$D(t) = S_0(t) - B(t) \tag{1}$$

The average values of clinical features that are absent in pregnant women, including blood pressure and oxygen levels, are substituted. The zero-mean approach is then used to standardize the clinical aspects of the mother. We use a model as outlined in [26] to extract high-level features from the preprocessed CTG signals and combine these features with the clinical data once the raw CTG features and the clinical data have been processed.

**2.3.2 GAN training.** BEGAN models are currently trained on datasets that were refined during the preprocessing phase. The generator in the BEGAN model has the same design as the AE, and the discriminator is made up of five layers. The dataset is separated into several classes before the BEGAN model is trained, and generative models are then created for each class. This means that after a generative model is fully trained, it will each produce synthetic data for the relevant class. Determining the criterion for ending the training process is a crucial step in using the BEGAN model to attack detection. Because it directly correlates with the caliber of synthetic data used to train the detection model, this choice has a significant impact on anomaly detection effectiveness. Fortunately for BEGAN, unlike other GAN models, it can estimate training convergence by using the equilibrium idea. Establishing the requirements for concluding the course is made easier by this feature. The following is the BEGAN convergence formula:

$$\mathbb{C}.\mathbb{M} = \lambda(a) + |\alpha\lambda(a) - \lambda(G(z))| \tag{2}$$

The reconstruction error function is represented by the term $\lambda(.)$ in Equation 2, whereas the diversity ratio for a given class of dataset is denoted by $\alpha$.

Using the convergence measure $M$ ends the training phase of a generative model. This means that throughout the training phase, the system input parameter must be treated as a threshold value. The training procedure is stopped if the convergence measure $M$ yields a value below the designated threshold. In the paradigm we propose, we set the $M$ threshold value as 0.058. The system uses the trained generator to create synthetic data based on the classes after the generative model has been trained. The initial training data is then combined with this created dataset. In the following step, the AE and the detection model are further trained using this expanded dataset. The synthetic data generation module was originally intended to produce as many generative models as there were classes; however, by utilizing

the idea of the conditional GAN architecture [36], in which class attributes are embedded within the input space, it can also be constructed as a single model.

**2.3.3 AE training.** The AE model must first be trained to carry out dimension reduction and feature extraction procedures in order to build an effective anomaly detection model. The generative model discriminator's design and the AE architecture in our suggested submodule are comparable. An AE model is constructed, trained on the enlarged dataset, and then the trained encoder is applied to the feature extraction process. Algorithm 1 represents AE's entire training procedure. It is noteworthy that the input layer of the detection models is the trained encoder, which is placed first. It is set up only to be a feature extractor, and it stays that way while the detection models are being trained.

**Algorithm 1** Training of Autoencoder with Generators

```
 1: INPUT : Training DataSet (TD) TD_train and set of generators G
 2: Initialization of AE parameters μ⁰_AE
 3: for G_i ∈ G, where 1 ≤ i ≤ k do
 4: z = {z_j}j = 1, 2, 3...., m_i
 5: Synthetic DataSet (SD) = G_i(z)
 6: end for
 7: Expanded Dataset (ED) = TD_train ∪ SD_1 ∪ ......... ∪ SD_k
 8: μ_AE = Train_AE(μ⁰_AE, ED)
 9: μ_E = Encoder(μ_AE)
10: OUTPUT: Trained Encoder (μ_E)
```

**2.3.4 Predictive model training.** At this point, we used Deep Neural Network (DNN), CNN, and LSTM—basic deep learning algorithms—to categorize abnormalities. Due to its inherent features, our DNN model, which has two hidden layers, performs well in identifying CTG anomalies using the supplied fine-tuned datasets. CNN is the second classifier, and it was created mainly to analyze picture datasets. To make it suitable for CTG data classification in terms of layers and input data space, a few structural modifications are needed. Hence, rather than transforming the input data into 2-D space, the CNN is built utilizing one-dimensional (1-D) convolutional layers for the classification of the CTG imbalance dataset. Consequently, the CNN model is comprised of one fully connected layer and two one-dimensional convolutional layers. Fig 7 shows the CNN's structural layout.

LSTM is the third classification DL model. It is represented in Fig 7 and consists of two recurrent layers with LSTM units and a fully connected layer. For the analysis of temporally linked characteristics, LSTM is especially helpful [37]. The output layer of all DL models has multivalued fields when the goal is to identify threat kinds in addition to detecting anomalies, and binary fields when the goal is to detect anomalies alone. Using the trained generators and encoder, Algorithm 2 describes the comprehensive method for training the predictive model. To put it briefly, the training of the detection model comes first, followed by data preparation, before the anomalies detection and classification model operates. We use $G - DNN_{AE}$, $G - CNN_{AE}$, and $G - LSTM$ to describe our model. Additionally, we have categorized the predictive models into three groups for a thorough comparison:

- CNN, DNN, and LSTM, which are referred to as naive DL models.
- $DNN_{AE}$ and $CNN_{AE}$, which are models combined with autoencoders and are called advanced DL models.
- $G - DNN_{AE}$, $G - CNN_{AE}$, and $G - LSTM$, which are the proposed models.

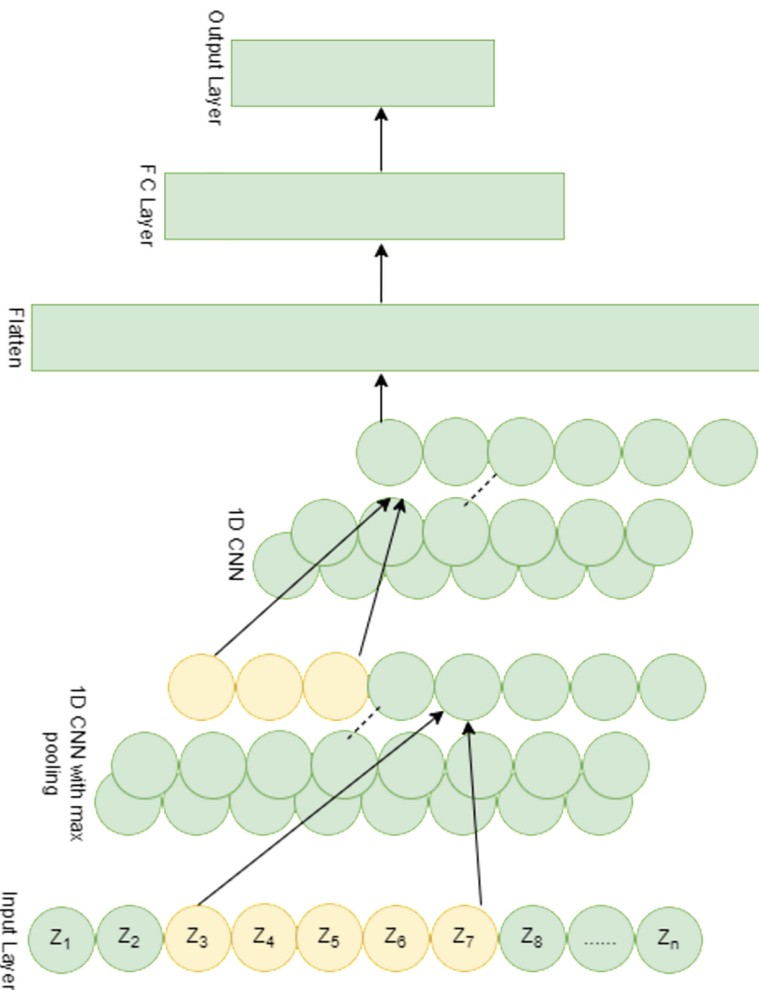

**Fig 7. The CNN classifier architecture used in our suggested framework to categorize CTG imbalance data into three groups: abnormal, non-abnormal, and reassuring.**

**Algorithm 2 Generators based classifier training.**

1: INPUT : Training DataSet (TD) $TD_{train}$, set of generators $\mathbf{G}$, and Trained Encoder $(\mu_E)$
2: Initialization of classifier parameters $\chi^0$
3: **for** $G_i \in \mathbf{G}$, where $1 \leq i \leq k$ **do**
4: $\mathbf{z} = \{z_j\} j = 1, 2, 3...., m_i$
5: Synthetic DataSet (SD) = $G_i(\mathbf{z})$
6: **end for**
7: Expanded Dataset (ED) = $TD_{train} \cup SD_1 \cup ......... \cup SD_k$
8: Trainable state of $\mu_E$ is set to false
9: Build $\chi^0_{\mu_E}$ = Model-Concatenation $(\mu_{AE}, \chi^0)$
10: $\chi_{\mu_E}$ = Train-Classifier $(\chi^0_{\mu_E}, ED)$
11: OUTPUT: Trained Encoder $(\chi_{\mu_E})$

## 2.4 Application layer

Medical specialists are shown the findings of anomaly detection and categorization in the application layer. The given data is divided into three classes in our framework: 1) Typical; 2) Doubting; and 3) Irregular. Instead, we use binary classification, which merges the "Pathological" and "Suspicious" classifications into a single "Abnormal" class. 'Suspicious' and 'Pathological' situations are thus represented by the 'Abnormal' class. Gynecologists and pediatricians then review the patient's classification to decide on the next best step.

## 3 Experimentation and analysis

We will first go over the SDN-IoT based experimental setup in this section, which was utilized to gather a small dataset for the purpose of evaluating the proposed framework. The dataset will then be described in detail, together with the experimental findings, a comparative analysis, and an assessment of the proposed system.

## 3.1 Experimental setup

Using the SDN-IoMT imbalance dataset, we have developed a GAN-AE DL-based system for minor classes anomaly detection and classification in an SDN-IoT context, with an emphasis on CTG features for fetal health monitoring. After a thorough examination of the literature [38], it is clear that no publically dataset is currently available for SDN-IoMT to analyze and assess the efficacy of the proposed technique. There is just one dataset available for SDN-IoT environments in the given literature [38], and it is mostly focused on network traffic intrusion detection [39]. Furthermore, there is just one dataset available for the Internet of Healthcare Things (IoHT) [40]; however, this dataset does not take SDN integration into account and instead focuses solely on the IoT-Healthcare domain.

Since there is no imbalanced Medical IoT dataset available to evaluate our proposed framework in an SDN network setting. We created the SDN-IoT environment with the Mininet 2.3.0 simulation tool. We then implemented a DL model using the TensorFlow framework, which is based on Python, inside the ONOS SDN controller. TensorFlow v2.12.0, the most recent version, is installed in our setup. A laptop equipped with an 8th generation Intel Core i9 processor, 16 GB of RAM, and a 1TB hard drive was used to run the simulations. To gather clinical characteristics of pregnant mothers, we simulated four different types of medical sensors. Then, we used a CNN model to extract four high-level values: FHR, BV, Acceleration, and Deceleration. These values are critical for identifying and analyzing eight important aspects that are pertinent to monitoring fetal distress in a distributed setting. Table 2 provides a complete description of these significant features, the matching sensors, and their accompanying normal ranges.

## 3.2 Dataset description

Using the technique outlined in [39], we have produced an imbalanced dataset based on CTG and clinical features within the SDN-IoT ecosystem. We gathered 50,000 samples in all for the project. Of them, 42,000 were deemed reliable and appropriate for testing, with the remaining 8,000 instances being disregarded. A 7:3 ratio was used to divide the dataset into training and test data. Following that, an 8:2 ratio was used to further split the training data into training and validation data. Eight features make up this dataset, four of which are mother-related and include temperature, pulse, oxygen saturation percentage, and maternal blood pressure. The final four features—FHR, BV, acceleration, and deceleration—are CTG characteristics. The two most crucial characteristics among these eight are FHR and Deceleration. 96% of

**Table 2. A description of CTG features reassuring and non-reassuring values.**

| Sr.No | CTG Features | Reassuring Value | Non-Reassuring Value |
|---|---|---|---|
| 1 | Fetal Heart Rate | 110-160 | < 110 and > 160 |
| 2 | Baseline Variability | 5-25 | < 5 and > 25 |
| 3 | Deceleration | No Dec., Early dec., Variable dec. | Regular, Late dec., prolonged dec. |
| 4 | Acceleration | 2 within 20 minutes | < 2 or *Absent* |

the 42,000 cases fall into the normal class, while 4% fall into the abnormal class on average. Within the 'abnormal' class, 2.7% fall into the suspicious class and 1.3% into the pathological category. Table 2 provides the normal values for each of the eight attributes. When a case falls into the suspicious category, it means that at least one of the four CTG characteristics is non-reassuring. Similarly, a case belonging to the pathological category suggests that two out of the four CTG traits are non-reassuring. Table 3 displays the reassuring and non-reassuring values for the CTG features. The whole dataset distribution for the two classes—normal and abnormal—is shown in Table 4.

### 3.3 Execution and adjusting parameters

As stated before, this framework's main goal is to identify and categorize anomalies in unbalanced data that comes from a variety of diverse SDN-IoT devices. In this study, we gathered CTG unbalanced datasets associated with SDN-IoT and tested them with our proposed architecture. The GAN discriminator is constructed with three layers within the proposed architecture. There are fifty latent space dimensions and eighty neurons in the first hidden layer. Thus, 80 neurons with a 50-dimensional latent space are also included in the generator's hidden layer. ReLU is the activation function that is being used. Notably, the AE has the same architecture as the discriminator and acts as a feature extractor. Additionally, we determined that 0.058 was the GAN convergence threshold. If the model's epoch count drops below 280 or below this threshold, the training process comes to an end. Similarly, 300 epochs is the end of the AE training. We have chosen three models for classification: CNN, LSTM, and DNN, each with two hidden layers. The DNN has sixteen neurons in the second layer and thirty-two in the first. Our CNN uses two convolutional layers in a 1-D-CNN architecture. 32 convolutional filters make up the first layer, while 16 neurons make up the second, which is a fully connected layer. In the CNN, ReLU serves as the activation function. The final model we employ for categorization is LSTM, with 64 connected LSTM cells making up each layer. We also concatenate the LSTM with a fully linked layer comprising 32 neurons. We have divided the DL algorithms into three categories—naive DL models, advanced DL models, and the suggested GAN-based DL models—in order to make a thorough comparison and analysis easier. CNN, LSTM, and DNN are included in the first category. $DNN_{AE}$ and $CNN_{AE}$ make up the second category, and $G - LSTM$, $G - DNN_{AE}$, and $G - CNN_{AE}$ make up the third. Four metrics

**Table 3. CTG imbalance dataset distribution.**

| Class | Training | Weight% | Testing | Weight% |
|---|---|---|---|---|
| Normal | 27,930 | 95 % | 12,222 | 97% |
| Suspicious | 970 | 3.3% | 265 | 2.1% |
| Pathological | 500 | 1.7 % | 113 | 0.9% |
| **Total** | **29,400** | **100%** | **12,600** | **100%** |

**Table 4. Performance of classifiers for normal and abnormal task detection using Recall (R), Precision (P), and F1-score (F1). G.LSTM, G.DNN, and G.CNN denote proposed models with Noise-Aware Encoder (NAE).**

| Classifier | A | Normal | | | Abnormal | | |
|---|---|---|---|---|---|---|---|
| | | R | P | F1 | R | P | F1 |
| *DNN* | 80.6% | 97.3% | 68.8% | 80.7% | 68.9% | 97.3% | 80.7% |
| *CNN* | 81.6% | 97.6% | 69.8% | 81.4% | 70.6% | 97.7% | 81.9% |
| *LSTM* | 83.1% | 98.6% | 72.1% | 83.2% | 71.1% | 98.3% | 82.4% |
| *DNN$_{AE}$* | 86.6% | 98.9% | 79.1% | 88.3% | 73.6% | 98.8% | 84.6% |
| *CNN$_{AE}$* | 87.5% | 98.9% | 80.1% | 88.9% | 75.2% | 98.8% | 85.7% |
| *G.LSTM* | 86.6% | 98.8% | 79.4% | 88.3% | 73.6% | 98.9% | 84.6% |
| *G.DNN$_{AE}$* | 90.9% | 98.9% | 85.4% | 91.7% | 82.6% | 98.9% | 90.1% |
| *G.CNN$_{AE}$* | 91.4% | 98.3% | 86.4% | 92.9% | 84.6% | 97.9% | 90.8% |

**A** = Accuracy, **R** = Recall, **P** = Precision, **F1** = F1-score, *G.LSTM, G.DNN$_{AE}$, G.CNN$_{AE}$* = Proposed Framework

have been used to compare the DL models that are naive, advanced, and the suggested models for attack detection and classification: F1-score, Accuracy, Precision, and Recall.

- Accuracy:

$$Accuracy = \frac{TP + TN}{TP + FP + TN + FN} \tag{3}$$

- Precision:

$$Precision = \frac{TP}{TP + FP} \tag{4}$$

- Recall:

$$Recall = \frac{TP}{TP + FN} \tag{5}$$

- F1-score

$$F1 - score = 2 * \frac{Precision * Recall}{Precision + Recall} \tag{6}$$

In this case, TN stands for true negatives, FP for false positives, FN for false negatives, and TP for true positives. Using the experimental datasets, we assessed each model using these metrics. It is important to note that although we created the models with a strong framework, questions persisted regarding their consistency and durability. Therefore, we performed 100 independent training runs for each model in order to compare and evaluate them, and we published the results based on the model that achieved the best detection rate in the test dataset.

## 3.4 Experimentation on the CTG and clinical features dataset

We used both binary and multiclassification techniques to test our dataset. We used a 7:3 ratio to split the dataset into training and testing data. There are 28,224 rows in our training data and 12,096 rows in the testing data. We used the generative model to build synthetic data for each class in our system experiments involving (i.e., $G - DNN_{AE}, G - CNN_{AE}, G - LSTM$) and added them to the training dataset. To achieve fair comparisons, it's crucial to remember that every model evaluation was done using the original test dataset.

**Table 5. Evaluating the proposed framework's accuracy in binary and multiclassification scenarios against the baseline DL models.**

| Models | Binary Classification Accuracy(%) | Multiclassification Accuracy(%) |
|---|---|---|
| Naive DL Models | 83.1 | 83.7 |
| Advanced DL Models | 87.5 | 89.6 |
| Proposed Framework | 91.4 | 94.2 |

**3.4.1 Binary classification.** The binary classification results are shown in Table 5, where abnormal refers to the category that includes both pathological and suspicious cases. Interestingly, the proposed system produced 6,000 synthetic data points for each class by using a trained GAN. Refer to Figs 8, 9, and 10 for a comparison of outcomes between naïve DL models, advanced DL models, and the proposed framework using the imbalanced CTG dataset. The experimental results show that the precision value is highest in the abnormal class and the recall value is highest in the normal class. With an accuracy of 83.1%, LSTM performs better than other naive DL models. By comparison, with accuracies of 87.5% and 86.6%, respectively, the advanced DL models, $CNN_{AE}$ and $DNN_{AE}$, surpass the naive DL models. With its combination of AE and GAN, the proposed framework outperforms both basic and sophisticated DL models, with accuracy values of 90.9% for $G.DNN_{AE}$ and 91.4% for $G.CNN_{AE}$. Regarding the LSTM model, the GAN-based LSTM performs better than the naive LSTM but is not as accurate as $CNN_{AE}$. The recall, precision, and F1-score of the three models on normal and abnormal traffic are shown in Fig 9 and Fig 10, respectively. Recall, precision, and F1-score values of 98.3%, 86.4%, and 92.9%, respectively, were obtained by the proposed approach. It is clear from the numbers that the proposed framework performed better than both baseline models (naive and advanced). A comparison of the accuracy of the proposed

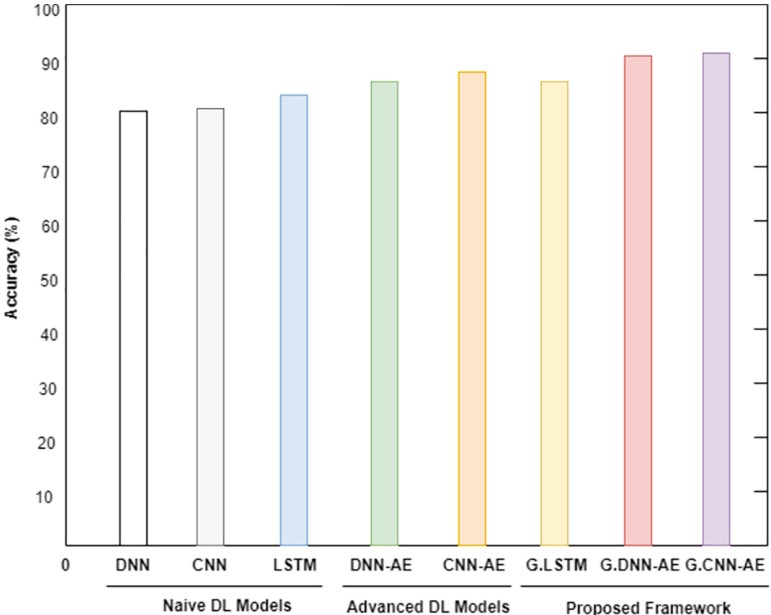

**Fig 8. Using the Accuracy metric, three model types—naive deep learning models, advanced deep learning models, and the proposed framework—are compared for the binary classification of normal and abnormal classes in an unbalanced CTG test dataset.**

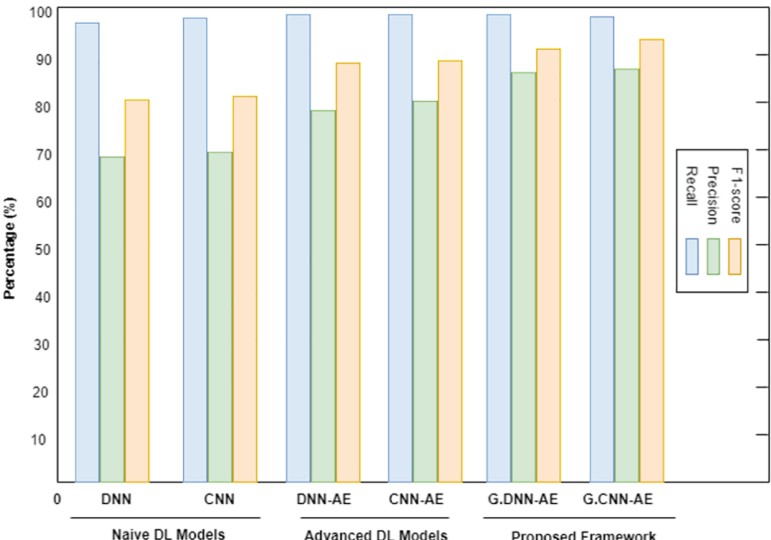

**Fig 9. Using the Recall, Precision, and F1-score metrics, three model types are compared: naïve deep learning models, advanced deep learning models, and the proposed framework for binary classification of normal class in an imbalanced CTG test dataset.**

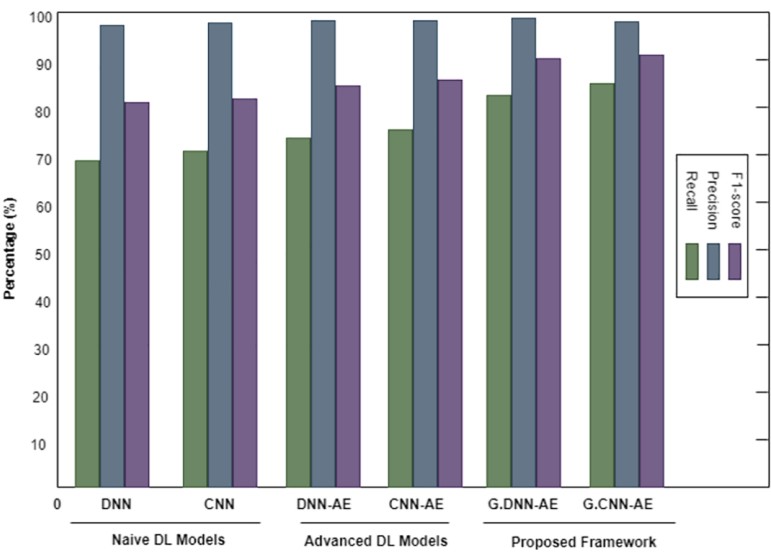

**Fig 10. Using the Recall, Precision, and F1-score metrics, three model types are compared: naïve deep learning models, advanced deep learning models, and the proposed framework for binary classification of abnormal class in an imbalanced CTG test dataset.**

framework in binary classification and multiclassification scenarios with the baseline DL models is shown in Table 6. Additionally, Fig 11 and Fig 12 present the confusion matrix and ROC analysis of the proposed framework on the imbalanced CTG dataset, respectively.

**3.4.2 Multiclassification.** The multiclassification findings for the imbalanced CTG dataset are shown in Table 7. The abnormal class is further split into several sub-classes in multiclassification. The abnormal class in our instance is separated into sub-classes that are suspect and pathological. For every subclass, synthetic data is produced according to

**Table 6. Performance comparison of classifiers for Suspicious and Pathological categories.**

| Classifier | A | Suspicious | | | Pathological | | |
|---|---|---|---|---|---|---|---|
| | | R | P | F1 | R | P | F1 |
| DNN | 80.7 % | 6.8% | 78.5% | 10.3% | 5.6% | 76.9% | 9.8% |
| CNN | 81.2 % | 8.3% | 83.2% | 13.4% | 7.3% | 81.8% | 12.5% |
| LSTM | 83.7 % | 8.1% | 80.4% | 12.3% | 6.3% | 78.9% | 11.9% |
| $DNN_{AE}$ | 89.4 % | 9.3% | 87.5% | 17.5% | 8.9% | 85.5% | 15.2% |
| $CNN_{AE}$ | 89.6 % | 12.3% | 88.3% | 19.3% | 10.4% | 87.7% | 17.8% |
| G.LSTM | 88.8 % | 12.9% | 85.9% | 21.6% | 11.3% | 83.8% | 19.1% |
| $G.DNN_{AE}$ | 93.3 % | 14.3% | 84.3% | 24.1% | 13.5% | 82.0% | 22.5% |
| $G.CNN_{AE}$ | 94.2 % | 13.9% | 85.8% | 23.6% | 12.6% | 82.0% | 21.1% |

**A** = Accuracy, **R** = Recall, **P** = Precision, **F1** = F1-score, **G.LSTM**, **G.DNN**$_{AE}$, **G.CNN**$_{AE}$ = Proposed Framework

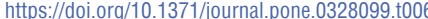
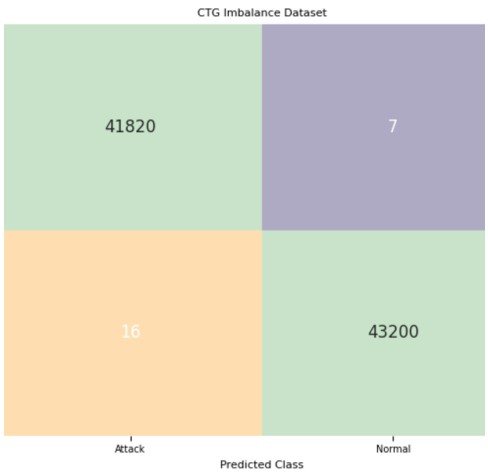

**Fig 11. The proposed framework's confusion matrix on the unbalanced CTG dataset.**

its relative importance in the population as a whole. The proposed approach produces AI data for minor classes that include less than 10% of the total population. Consequently, a trained generative model generates 10,000 synthetic data points for both the suspicious and pathological classifications. Using the CTG dataset, Figs 13, 14, and 15 show how experimental results compare in the multiclassification situation. Naive deep learning models perform similarly to binary classification when evaluated using the accuracy metric shown in Fig 13. It is noteworthy, nonetheless, that naive DL models outperform fundamental machine learning approaches like SVM and DT for minor classes like the suspicious and pathological classes, with F1-Score values exceeding those of the former. The advanced DL models $DNN_{AE}$ and $CNN_{AE}$ have accuracy values of 89.4% and 89.6%, respectively. All things considered, these models outperform the naive DL models in terms of classification performance. Nevertheless, advanced models did not show a statistically significant increase when we specifically compared the classification performance of advanced DL models to naive DL models for the extremely minor class, such as the pathological class. In contrast, the proposed architecture used $G - DNN_{AE}$ and $G - CNN_{AE}$ to reach remarkable accuracy values of 93.3% and 94.2%, respectively. The performance of the proposed method was significantly improved when viewed from the viewpoint of minor classes, exceeding both simple and complex DL models. The three metrics—recall, accuracy, and F1-score—show that the proposed framework outperforms the naïve and advanced DL models in Figs 14 and 15. Furthermore, the

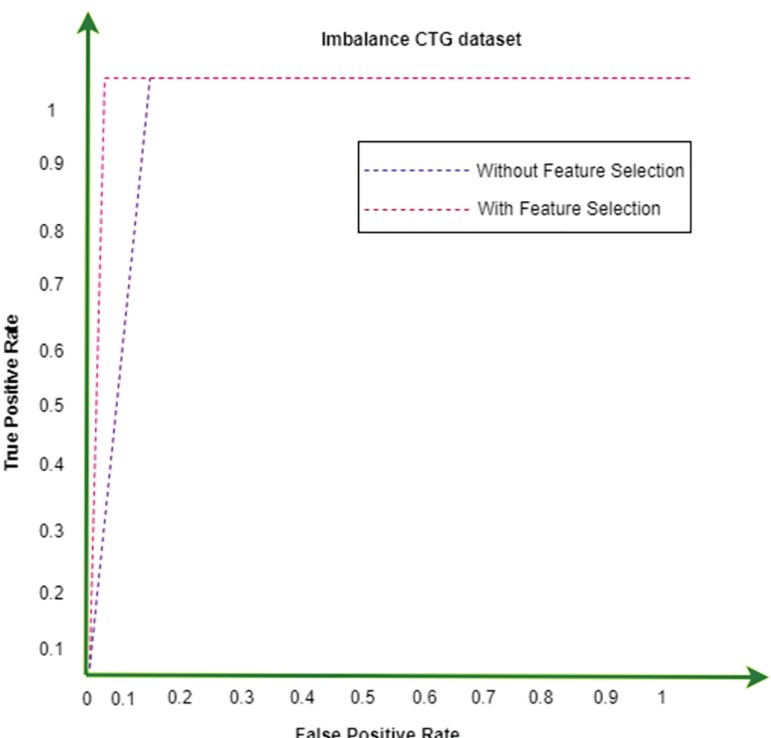

**Fig 12. ROC analysis of the proposed framework on the imabalnace CTG dataset.**

**Table 7. Performance metrics with mean, standard deviation, and 95% confidence interval.**

| Metric | Mean (%) | Standard Deviation (±) | 95% Confidence Interval (%) |
|---|---|---|---|
| Accuracy | 91.4 | ±1.2 | [90.9–91.9] |
| Precision | 86.4 | ±1.5 | [85.9–86.9] |
| Recall | 98.3 | ±0.9 | [98.0–98.6] |
| F1-Score | 92.9 | ±1.1 | [92.5–93.3] |

testing time of the proposed model is illustrated in Fig 16. It is evident from Fig 16 that the proposed model outperformed both baseline deep learning models in terms of speed efficiency, achieving a testing time value of 19.75 ms. Conversely, $DNN_{AE}$ demonstrated superior testing time compared to the LSTM model.

In conclusion, the AE model combined with CNN and DNN models has significantly improved classification accuracy in binary and multiclassification scenarios. Furthermore, adding the GAN model can increase this increased accuracy even further.

## 4 Results and discussion

This section describes the outcomes that the proposed GAN-AE based framework produced when it was used to analyze imbalanced CTG dataset in a simulated SDN-IoMT scenario. Detailed testing shows that the framework outperforms both simple and complex DL models in terms of identifying and categorizing small class abnormalities. The Generative Adversarial Network's ability to produce plausible synthetic data for minor classes, which effectively addresses the problem of data imbalance, is credited with the impressive performance on unbalanced data. More specifically, the "Suspicious" and "Pathological" minor classes showed

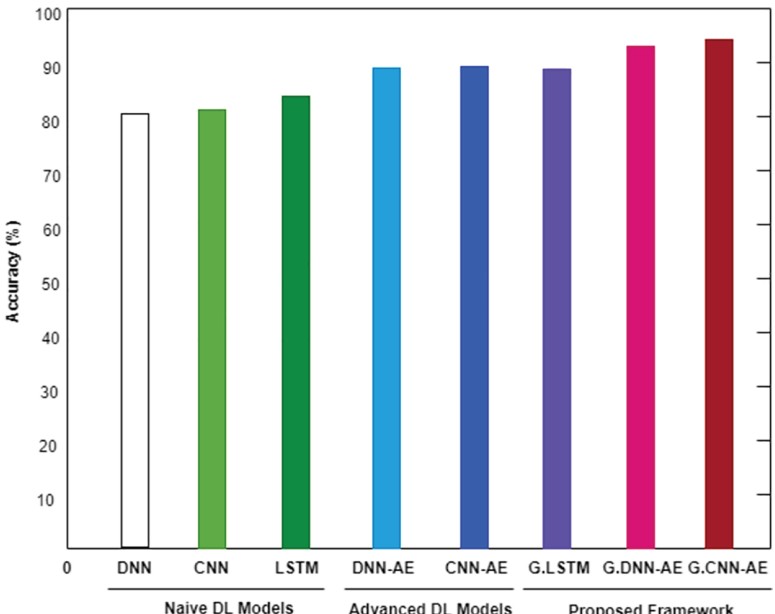

**Fig 13. Three types of models are compared using the Accuracy metric for multiclassification in an imbalanced CTG test dataset: basic deep learning models, advanced deep learning models, and the proposed framework.**

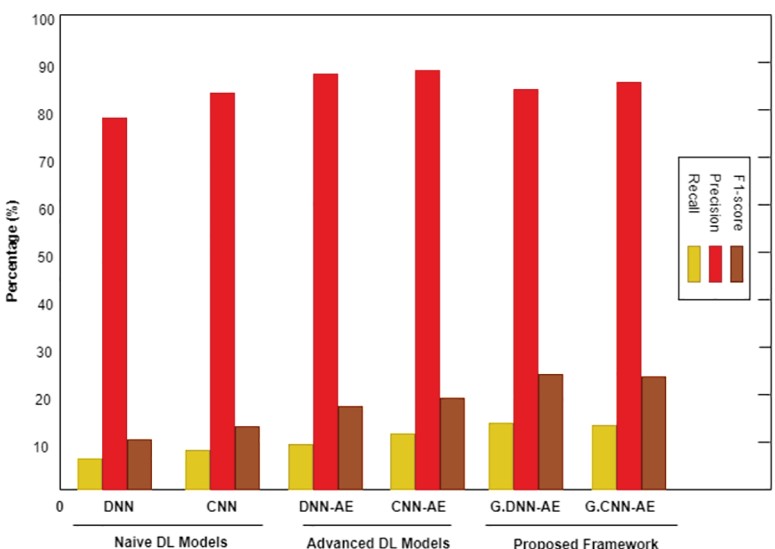

**Fig 14. Using the Recall, Precision, and F1-score metrics, three model types are compared: naïve deep learning models, advanced deep learning models, and the proposed framework for multiclassification of Suspicious minor class in an imbalanced CTG test dataset.**

a notable performance improvement with the proposed models. This method outperformed both naive and sophisticated DL models, with accuracy rates in binary classification and multiclassification of 91.4% and 94.2% , respectively. Figs 8 and 10 provide comprehensive results. Additionally, using the publicly available CTG dataset CTU-UHB, the proposed framework's performance was contrasted with the most recent methods found in the literature, as shown

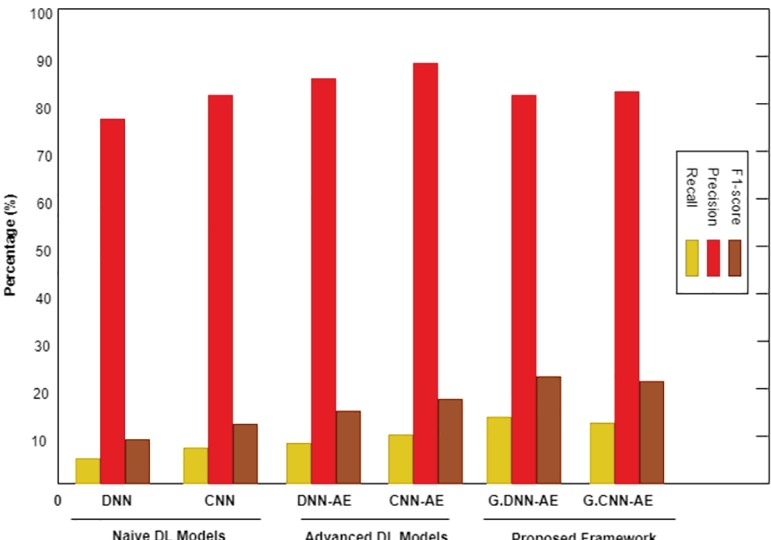

**Fig 15. Using the Recall, Precision, and F1-score metrics, three model types—naive deep learning models, advanced deep learning models, and the proposed framework—are compared for multiclassification of the Pathological minor class in an imbalanced CTG test dataset.** The Proposed Framework is indicated here by P.F.

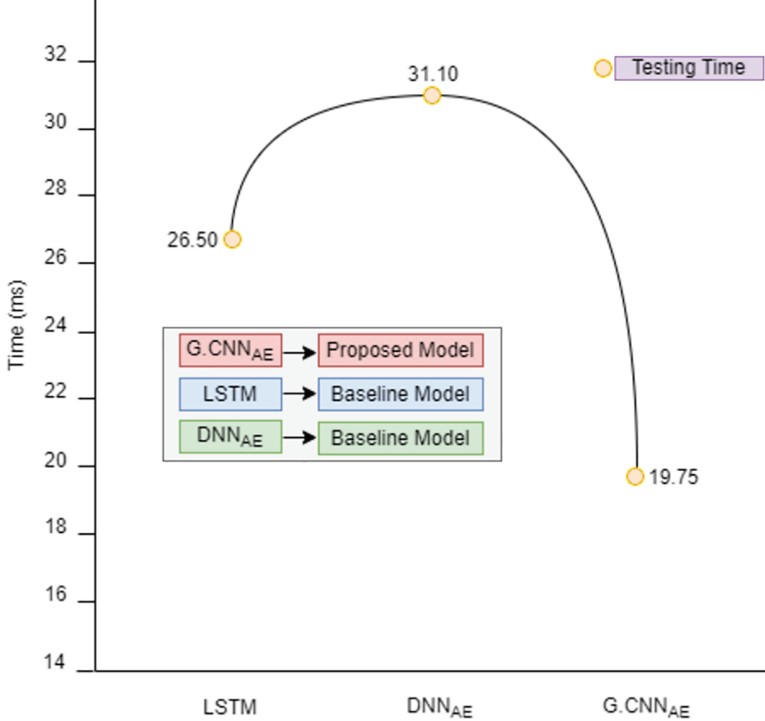

**Fig 16. Speed efficiency of the proposed technique in comparison with the baseline deep learning models.**

in Table 8. The outcomes demonstrate the promising performance of the framework, attaining an accuracy of 94.6%, which is the highest of all existing methods for handling imbalanced

**Table 8. Statistical comparison of model performance.**

| Comparison | Test Type | Metric Compared | p-value / 95% CI |
|---|---|---|---|
| GAN-AE vs. CNN (Binary) | McNemar's Test | Accuracy | p = 0.003 |
| GAN-AE vs. LSTM (Binary) | McNemar's Test | Accuracy | p = 0.011 |
| GAN-AE vs. AE (Multiclass) | Bootstrap CI | F1-score diff | [1.5%, 3.2%] |
| GAN-AE vs. CNN (Multiclass) | Bootstrap CI | Accuracy diff | [1.1%, 2.6%] |

CTG data. We proved the efficacy of the framework in real-world scenarios by demonstrating through experiments on the CTU-UHB dataset that it can greatly improve the detection performance of small class abnormalities by resolving data imbalance. We conducted 100 iterations of each experiment to evaluate the stability of the suggested GAN-AE-based SDN-IoMT anomaly detection model. The average performance, standard deviations, and 95% confidence intervals are summarized in Table 7 . The average accuracy, precision, recall, and F1-score of the model were 91.4% ± 1.2, 86.4% ± 1.5, and 98.3% ± 0.9, respectively. The statistical dependability and consistency of the model's performance over several runs are validated by these small confidence intervals.

We used rigorous statistical testing to confirm that the suggested GAN-AE model's performance gains over baseline models are statistically significant. We compared the prediction results of the GAN-AE model with each baseline model (e.g., CNN, LSTM, autoencoder without GAN augmentation) for the binary classification problem using McNemar's test are summarized in Table 8. All of the p-values that were found were less than 0.05, which suggests that the categorization performance varied statistically significantly. We used a bootstrap resampling technique (10,000 resamples) to calculate 95% confidence intervals for the difference between the suggested model and baselines in terms of F1-score and overall accuracy for the multiclass classification situation. As shown in Table 9, the proposed framework achieves the highest accuracy of 94.60% on the CTU-UHB dataset. The statistical significance of the observed improvements was supported by the fact that 0 was not included in the calculated confidence ranges.

## 5 Clinical scope and intended application window

The proposed SDN-IoMT deep learning framework has been developed and validated using antepartum cardiotocography (CTG) data, primarily focused on remote monitoring of maternal and fetal health during the third trimester before the onset of labor. This period is

**Table 9. Comparison of the proposed framework with the existing literature.**

| Ref. | Methods | Dataset | Accuracy(%) |
|---|---|---|---|
| [21] | TS-GAN | CTU-UHB | 71.08% |
| [22] | IBTF | CTU-UHB | 64.01% |
| [23] | FCN | CTU-UHB | 50.21% |
| [24] | GNB | CTU-UHB | 65.91% |
| [25] | RF | CTU-UHB | 90.80% |
| [26] | MMDLA | CTU-UHB | 90.77% |
| [28] | Alexnet | CTU-UHB | 93.20% |
| [30] | GCB | CTU-UHB | 93.90% |
| [31] | SLP | CTU-UHB | 94.00% |
| P.F | GAN-AE | CTU-UHB | 94.60% |
| P.F = Proposed Framework | | | |

characterized by relatively stable physiological patterns with minimal external stressors such as uterine contractions or acute fetal distress, making it well suited for early anomaly detection and continuous risk assessment. However, intrapartum monitoring—which occurs during active labor—introduces additional complexity. Uterine activity, contraction-induced stress, and dynamic changes in fetal heart rate (FHR) variability during labor can significantly alter signal morphology. These conditions may result in non-stationary patterns and abrupt signal transitions that were not present in the antepartum training data. As such, while the current model demonstrates high performance in the antepartum context, its direct application to intrapartum scenarios is not yet validated. For clinical deployment in labor monitoring settings, it would be essential to:

- Retrain or fine-tune the model using labeled intrapartum data that captures stress-related signal variations.
- Incorporate additional features such as uterine contraction frequency, intensity, and fetal response latency to enhance sensitivity to labor-related anomalies.

This limitation has been acknowledged in the current version of the manuscript, and the authors identify adaptation to intrapartum monitoring as a key direction for future research and clinical translation.

## 6 Conclusion

An AI-driven framework designed for fetal distress monitoring in an SDN-IoMT setting is presented in this study. SDN specifically coordinates, in a distributed architecture, the management and control of a wide variety of IoMT devices. The CTG data imbalance issue is well resolved by the proposed GAN-AE framework, which also improves detection and classification performance. We used two state-of-the-art deep learning models, AE and GAN, that are meant to produce plausible synthetic data in order to address the problem of data imbalance. This method was rigorously tested and analyzed on the open-access CTU-UHB CTG dataset as well as an imbalanced CTG dataset. We evaluated the suggested framework using accuracy, precision, recall, and F1-score and found that it performed better than baseline deep learning models. Applying the suggested framework to the imbalanced CTG dataset, it achieved a remarkable accuracy of 91.4% in binary classification and 94.2% in multiclassification. It was noteworthy for its outstanding performance, especially in the very small classes—that is, the "suspicious" and "pathological" classes. Additionally, a comparative study using the publicly available CTU-UHB dataset was carried out in relation to a number of the most recent methods reported in the literature.

## Author contributions

**Conceptualization:** Amin Ullah, Qazi Mazhar ul Haq, Zabeeh Ullah.

**Data curation:** Amin Ullah, Qazi Mazhar ul Haq, Zabeeh Ullah.

**Formal analysis:** Zabeeh Ullah, Muhammad Shahid Anwar.

**Funding acquisition:** Jaroslav Frnda, Muhammad Shahid Anwar.

**Investigation:** Qazi Mazhar ul Haq, Jaroslav Frnda, Muhammad Shahid Anwar.

**Methodology:** Amin Ullah, Qazi Mazhar ul Haq, Zabeeh Ullah.

**Project administration:** Qazi Mazhar ul Haq, Jaroslav Frnda.

**Resources:** Qazi Mazhar ul Haq, Zabeeh Ullah.

**Software:** Amin Ullah, Zabeeh Ullah.

**Supervision:** Qazi Mazhar ul Haq.

**Validation:** Amin Ullah, Qazi Mazhar ul Haq, Jaroslav Frnda, Muhammad Shahid Anwar.

**Visualization:** Qazi Mazhar ul Haq, Muhammad Shahid Anwar.

**Writing – original draft:** Amin Ullah, Qazi Mazhar ul Haq.

**Writing – review & editing:** Qazi Mazhar ul Haq.

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
