## [Decision Letter · Decision Letter 0]

19 Dec 2024

PONE-D-24-48368AI-Driven Fetal Distress Monitoring in AI-Native 6G Networks: An SDN-IoMT Framework for Enhanced Medical ConnectivityPLOS ONE

Dear Dr. Haq,

Thank you for submitting your manuscript to PLOS ONE. After careful consideration, we feel that it has merit but does not fully meet PLOS ONE’s publication criteria as it currently stands. Therefore, we invite you to submit a revised version of the manuscript that addresses the points raised during the review process.

We look forward to receiving your revised manuscript.

Kind regards,

Agnese Sbrollini

Academic Editor

PLOS ONE

Journal Requirements: When submitting your revision, we need you to address these additional requirements. 1. Please ensure that your manuscript meets PLOS ONE's style requirements, including those for file naming. The PLOS ONE style templates can be found at https://journals.plos.org/plosone/s/file?id=wjVg/PLOSOne_formatting_sample_main_body.pdf and https://journals.plos.org/plosone/s/file?id=ba62/PLOSOne_formatting_sample_title_authors_affiliations.pdf 2. Please note that PLOS ONE has specific guidelines on code sharing for submissions in which author-generated code underpins the findings in the manuscript. In these cases, we expect all author-generated code to be made available without restrictions upon publication of the work. Please review our guidelines at https://journals.plos.org/plosone/s/materials-and-software-sharing#loc-sharing-code and ensure that your code is shared in a way that follows best practice and facilitates reproducibility and reuse. 3. Thank you for stating the following financial disclosure: "This research was supported by the institutional research of the Faculty of Operation and Economics of Transport and Communications—University of Zilina, no. 3/FPEDAS/2024." Please state what role the funders took in the study.  If the funders had no role, please state: ""The funders had no role in study design, data collection and analysis, decision to publish, or preparation of the manuscript."" If this statement is not correct you must amend it as needed. Please include this amended Role of Funder statement in your cover letter; we will change the online submission form on your behalf. 4. Thank you for stating the following in the Acknowledgments Section of your manuscript: "This research was supported by the institutional research of the Faculty of Operation and Economics of Transport and Communications—University of Zilina, no.3/FPEDAS/2024" We note that you have provided funding information that is not currently declared in your Funding Statement. However, funding information should not appear in the Acknowledgments section or other areas of your manuscript. We will only publish funding information present in the Funding Statement section of the online submission form. Please remove any funding-related text from the manuscript and let us know how you would like to update your Funding Statement. Currently, your Funding Statement reads as follows: "This research was supported by the institutional research of the Faculty of Operation and Economics of Transport and Communications—University of Zilina, no. 3/FPEDAS/2024." Please include your amended statements within your cover letter; we will change the online submission form on your behalf. 5. We note that your Data Availability Statement is currently as follows: All relevant data are within the manuscript and its Supporting Information files. Please confirm at this time whether or not your submission contains all raw data required to replicate the results of your study. Authors must share the “minimal data set” for their submission. PLOS defines the minimal data set to consist of the data required to replicate all study findings reported in the article, as well as related metadata and methods (https://journals.plos.org/plosone/s/data-availability#loc-minimal-data-set-definition). For example, authors should submit the following data: - The values behind the means, standard deviations and other measures reported;- The values used to build graphs;- The points extracted from images for analysis. Authors do not need to submit their entire data set if only a portion of the data was used in the reported study. If your submission does not contain these data, please either upload them as Supporting Information files or deposit them to a stable, public repository and provide us with the relevant URLs, DOIs, or accession numbers. For a list of recommended repositories, please see https://journals.plos.org/plosone/s/recommended-repositories. If there are ethical or legal restrictions on sharing a de-identified data set, please explain them in detail (e.g., data contain potentially sensitive information, data are owned by a third-party organization, etc.) and who has imposed them (e.g., an ethics committee). Please also provide contact information for a data access committee, ethics committee, or other institutional body to which data requests may be sent. If data are owned by a third party, please indicate how others may request data access. 6. PLOS requires an ORCID iD for the corresponding author in Editorial Manager on papers submitted after December 6th, 2016. Please ensure that you have an ORCID iD and that it is validated in Editorial Manager. To do this, go to ‘Update my Information’ (in the upper left-hand corner of the main menu), and click on the Fetch/Validate link next to the ORCID field. This will take you to the ORCID site and allow you to create a new iD or authenticate a pre-existing iD in Editorial Manager.

Reviewers' comments:

Reviewer's Responses to Questions

**Comments to the Author**

1. Is the manuscript technically sound, and do the data support the conclusions?

Reviewer #1: No

Reviewer #2: Yes

2. Has the statistical analysis been performed appropriately and rigorously? 

Reviewer #1: N/A

Reviewer #2: Yes

3. Have the authors made all data underlying the findings in their manuscript fully available?

Reviewer #1: Yes

Reviewer #2: Yes

4. Is the manuscript presented in an intelligible fashion and written in standard English?

Reviewer #1: No

Reviewer #2: Yes

5. Review Comments to the Author

Reviewer #1: The paper describes the implementation of SDN and DL for fetal data streaming and classification. In the current form, it is very unclear the main contribution of the paper. It seems to tackle too many complex problems and not one in detail. For example, many information about the 6G network and SDN are provided but no formal tests have been reported. A Generative model is said to be developed for handling data imbalance, but no results on the quality of the generated data are reported. Finally, DL models are compared to determine fetal distress but i) no clear information about the gold standard are reported; ii) performance metrics not clear; and iii) model selection performed on the test set, which is a well-known methodological mistake. In addition, the paper requires extensive revision for clarity. Please find below other concerns.

-Not clear who and how were defined the categories for fetal distress.

-The performance metrics reported are suitable for binary classification. However, the problem tackled was a 3-class classification problem. Not sure authors computed exactly these quantitities. At a certain point, the problem becomes a binary one. Not clear at all.

-What is the objective of the authors with "we published the results based on the model that achieved the best detection rate in the test dataset"? Doing so, the test set becomes another validation set because selection is performed on these data. It is a methodological mistake if not well motivated.

-Figure 1 is never mentioned in the text.

-Figure 2 shows many CTG devices connected and able to stream their data. However, I am not aware of any commercial device which can do that.

-Please check the instruction for authors since the table of acronyms is rarely allowed by journals. Also, check all acronyms are instantiated once and in the right order (e.g. DL is used before instantiation)

-What is BV?

-Are authors sure that the GAN proposed in [21] is called "Reconstruction Error"?

-What is FHR's decentralization?

Reviewer #2: 1. The Sensing layer described is only meant to detect fetal cardiac activity and some maternal vital signs. It does not

detect uterine contractions. While fetal cardiac activity is interpreted based on uterine contraction. The sensing

should be able or developed to detect uterine contractions as well.

2. Why did was median used against mean to replace the missing data dst and bst?

3. Blood pressure (BP) of 139/99 is not normal (Table 2). Any diastolic BP greater that 89mmHg is High. With this AI

devise women with hypertensive disorders of pregnancy may be missed in the antenatal or labour period.

4. For more information on Blood pressure in pregnancy, check guidelines by International Society for the Study of Hypertension in Pregnancy and FIGO.

5. Other comments and corrections are in the text

6. PLOS authors have the option to publish the peer review history of their article (what does this mean?). If published, this will include your full peer review and any attached files.

Reviewer #1: No

Reviewer #2: **Yes: **Dr Bilal Sulaiman

---

## [Author Response · Author response to Decision Letter 1]

27 Feb 2025

Please check the attached files respond to reviewers

---

## [Decision Letter · Decision Letter 1]

4 May 2025

PONE-D-24-48368R1AI-Driven Fetal Distress Monitoring SDN-IoMT NetworksPLOS ONE

Dear Dr. Haq,

Thank you for submitting your manuscript to PLOS ONE. After careful consideration, we feel that it has merit but does not fully meet PLOS ONE’s publication criteria as it currently stands. Therefore, we invite you to submit a revised version of the manuscript that addresses the points raised during the review process.

We look forward to receiving your revised manuscript.

Kind regards,

Agnese Sbrollini

Academic Editor

PLOS ONE

Reviewers' comments:

Reviewer's Responses to Questions

**Comments to the Author**

1. If the authors have adequately addressed your comments raised in a previous round of review and you feel that this manuscript is now acceptable for publication, you may indicate that here to bypass the “Comments to the Author” section, enter your conflict of interest statement in the “Confidential to Editor” section, and submit your "Accept" recommendation.

Reviewer #2: (No Response)

Reviewer #3: All comments have been addressed

2. Is the manuscript technically sound, and do the data support the conclusions?

Reviewer #2: Yes

Reviewer #3: Yes

3. Has the statistical analysis been performed appropriately and rigorously? 

Reviewer #2: Yes

Reviewer #3: Yes

4. Have the authors made all data underlying the findings in their manuscript fully available?

Reviewer #2: Yes

Reviewer #3: No

5. Is the manuscript presented in an intelligible fashion and written in standard English?

Reviewer #2: Yes

Reviewer #3: Yes

6. Review Comments to the Author

Reviewer #2: 1. The revised manuscript neither have highlight nor tract changes to indicated changes made on it. This made it so difficult to identify changes made on the article.

2. There is no accompanying document explaining the responses line by line made by the reviewers.

Reviewer #3: Dear Authors,

Thank you for the opportunity to review your revised manuscript. Overall, you have strengthened the technical presentation and addressed many of the points from the previous round. Below, I summarize my key comments and remaining concerns:

Technical Rigor & Data Support

Your SDN-IoMT simulation using Mininet/ONOS and the GAN-AE models is well described, and your reported performance metrics (Accuracy, Precision, Recall, F1-score) are supported by the underlying data.

Repeating each experiment 100 times improves confidence in stability, and your binary and multiclass comparisons clearly demonstrate the benefit of GAN-based augmentation.

Statistical Analysis

Please include measures of variability (e.g., standard deviations or 95% confidence intervals) for your key metrics across the 100 runs.

Consider a formal paired statistical test (e.g., McNemar’s test or bootstrap confidence intervals) when comparing your GAN-AE models against baseline models to demonstrate significance.

Data Availability

While the CTU-UHB public dataset is fully accessible, the synthetic and clinical data generated in your SDN-IoMT simulations remain unpublished. Please deposit these source files (simulation scripts, anonymized clinical recordings, and any synthesized data) in a public repository or provide clear instructions for controlled access.

If data sharing is constrained by patient confidentiality, describe the anonymization process and specify how qualified researchers can obtain full datasets under an appropriate agreement.

CTG Scoring & Clinical Context

Clarify the intended application window: your framework has been trained on antepartum data. If you intend it for intrapartum (labor) monitoring, please discuss how uterine activity and labor stress patterns might affect performance.

Although you define custom thresholds for FHR, variability, accelerations, and decelerations, please justify these relative to established clinical guidelines (Fisher/Krebs, FIGO, or Dawes-Redman). If STV/LTV or sinusoidal pattern detection was not feasible, explain why and discuss the potential impact on detection of rare but critical pathologies.

Minor Points

In Table 1, please add Celsius equivalents for body temperature (36.1–37.2 °C).

Correct minor language issues (e.g., “heterogeneity” vs. “heterogeneous”) and ensure consistent figure axis labels.

Once these points are addressed, I believe your manuscript will be suitable for publication.

Best regards,

Yuliia Balycheva

7. PLOS authors have the option to publish the peer review history of their article (what does this mean?). If published, this will include your full peer review and any attached files.

Reviewer #2: **Yes: **Bilal Sulaiman

Reviewer #3: No

---

## [Author Response · Author response to Decision Letter 2]

31 May 2025

Please check the attachment for responses to reviewers comments

---

## [Decision Letter · Decision Letter 2]

26 Jun 2025

AI-Driven Fetal Distress Monitoring SDN-IoMT Networks

PONE-D-24-48368R2

Dear Dr. Haq,

We’re pleased to inform you that your manuscript has been judged scientifically suitable for publication and will be formally accepted for publication once it meets all outstanding technical requirements.

Kind regards,

Agnese Sbrollini

Academic Editor

PLOS ONE

Additional Editor Comments (optional):

Reviewers' comments:

Reviewer's Responses to Questions

**Comments to the Author**

1. If the authors have adequately addressed your comments raised in a previous round of review and you feel that this manuscript is now acceptable for publication, you may indicate that here to bypass the “Comments to the Author” section, enter your conflict of interest statement in the “Confidential to Editor” section, and submit your "Accept" recommendation.

Reviewer #3: All comments have been addressed

2. Is the manuscript technically sound, and do the data support the conclusions?

Reviewer #3: Yes

3. Has the statistical analysis been performed appropriately and rigorously? 

Reviewer #3: Yes

4. Have the authors made all data underlying the findings in their manuscript fully available?

Reviewer #3: Yes

5. Is the manuscript presented in an intelligible fashion and written in standard English?

Reviewer #3: Yes

6. Review Comments to the Author

Reviewer #3: The authors have adequately addressed all major comments from the previous review round. In particular, they clarified the clinical rationale for their CTG classification thresholds by aligning them with FIGO and Fisher/Krebs standards, and they explained the justified exclusion of STV, LTV, and sinusoidal patterns due to dataset limitations. Statistical validation has been improved with confidence intervals and appropriate significance testing. The revised manuscript is technically sound, clinically relevant, and ready for publication.

7. PLOS authors have the option to publish the peer review history of their article (what does this mean?). If published, this will include your full peer review and any attached files.

Reviewer #3: No

---

## [Editor Report · Acceptance letter]

PONE-D-24-48368R2

PLOS ONE

Dear Dr. Haq,

I'm pleased to inform you that your manuscript has been deemed suitable for publication in PLOS ONE. Congratulations! Your manuscript is now being handed over to our production team.

Kind regards,

on behalf of

Dr. Agnese Sbrollini

Academic Editor

PLOS ONE